# Integration of GWAS and RNA-Seq Analysis to Identify SNPs and Candidate Genes Associated with Alkali Stress Tolerance at the Germination Stage in Mung Bean

**DOI:** 10.3390/genes14061294

**Published:** 2023-06-19

**Authors:** Ning Xu, Bingru Chen, Yuxin Cheng, Yufei Su, Mengyuan Song, Rongqiu Guo, Minghai Wang, Kunpeng Deng, Tianjiao Lan, Shuying Bao, Guifang Wang, Zhongxiao Guo, Lihe Yu

**Affiliations:** 1College of Agriculture, Heilongjiang Bayi Agricultural University, Daqing 163319, China; xunig2008@163.com; 2Institute of Crop Germplasm Resources, Jilin Academy of Agricultural Sciences, Gongzhuling 136100, China; chenbingxu2008@126.com (B.C.); apple9036@163.com (Y.C.); s1694303087@163.com (Y.S.); smyuan2023@163.com (M.S.); 15243148967@163.com (R.G.); shiyongdou@163.com (M.W.); dkpeng0512@163.com (K.D.); lan15500180015@163.com (T.L.); bshy5015@163.com (S.B.); wangguifang2008@126.com (G.W.)

**Keywords:** mung bean, alkali tolerance, genome-wide association study, RNA-seq, candidate genes, germination stage

## Abstract

Soil salt-alkalization seriously impacts crop growth and productivity worldwide. Breeding and applying tolerant varieties is the most economical and effective way to address soil alkalization. However, genetic resources for breeders to improve alkali tolerance are limited in mung bean. Here, a genome-wide association study (GWAS) was performed to detect alkali-tolerant genetic loci and candidate genes in 277 mung bean accessions during germination. Using the relative values of two germination traits, 19 QTLs containing 32 SNPs significantly associated with alkali tolerance on nine chromosomes were identified, and they explained 3.6 to 14.6% of the phenotypic variance. Moreover, 691 candidate genes were mined within the LD intervals containing significant trait-associated SNPs. Transcriptome sequencing of alkali-tolerant accession 132–346 under alkali and control conditions after 24 h of treatment was conducted, and 2565 DEGs were identified. An integrated analysis of the GWAS and DEGs revealed six hub genes involved in alkali tolerance responses. Moreover, the expression of hub genes was further validated by qRT-PCR. These findings improve our understanding of the molecular mechanism of alkali stress tolerance and provide potential resources (SNPs and genes) for the genetic improvement of alkali tolerance in mung bean.

## 1. Introduction

Among several abiotic stresses, soil salt-alkalization is an increasingly severe global problem, as saline–alkali stress can negatively affect germination, seedling growth and development, vigor, and yield, which are huge concerns for crop breeders [1,2,3]. It is estimated that more than 800 million ha of land is damaged due to salt-alkalization globally, and unfortunately, 32 million ha of agricultural dryland is considered to be threatened [3]. Superficially, soil salt-alkalization means soil salinization and alkalization, which are, in fact, simultaneous processes [4]. Soil salinization results in salt stress caused by neutral salts, such as NaCl and Na_2_SO_4_, while soil alkalization leads to alkali stress caused by alkaline salts, such as Na_2_CO_3_ and NaHCO_3_, which can cause greater damage than neutral salts through the elevation of soil pH [5], in addition to causing osmotic stress and ion toxicity [6]. The mung bean (*Vigna radiata* L.) is one of the most important pulse crops. It is grown mainly in Asia, with a trend of becoming popular in other parts of the world due to its conspicuous virtues, e.g., as a nutrient source and medicinal component for humans, a forage crop for animals, and a source of soil fertilization and insect prevention for agroecological systems [7,8]. Furthermore, by relying on their wide adaptability and low input requirements [9], mung bean in China is largely planted on poor-quality land, especially land experiencing drought or saline–alkali conditions; over 40% of the production occurs in northeast China, which has 3.73 million ha of sodic land [10]. Mung bean, which is sensitive to saline–alkali stress, can be affected more heavily than other crops [11,12]. In general, there is an important practical significance for the development of tolerant varieties to saline–alkali land for mung bean production and the effective use of saline–alkali land.

Salinity–alkalinity tolerance in crops is a very complex quantitative trait controlled by multiple quantitative trait loci (QTLs) [13,14,15]. Identifying the QTLs for salinity–alkalinity-tolerance-related traits and developing molecular markers linked to causal genes as a selection tool for breeding have provided an efficient approach for reducing the losses caused by the saline–alkali threat [16,17]. Compared to traditional QTL linkage mapping using biparental populations, a genome-wide association study (GWAS) utilizes natural populations with abundant variation and can pinpoint multiple alleles that have accumulated in natural populations during evolution [18]. Limited loci associated with salt tolerance were detected in mung bean by the GWAS. Five SNPs related to salt stress (50 mM NaCl) tolerance were identified for the germination rate in two QTLs containing three salt-stress-tolerance-related genes (*Vradi07g01630*, *Vradi09g09510*, and *Vradi09g09600*) [19]. At the seedling stage, three and four significant QTLs were detected for the survival rate 10 days and 15 days after a salt treatment (100 mM NaCl), respectively, and the candidate gene *VrFRO8* could lower the SOD content by influencing the Fe^2+^/Fe^3+^ ratio under salt stress [20]. However, several hub genes related to the alkali response in other crops, such as *LOC_Os03g26210* [21], *LOC_Os11g37300*, *LOC_Os11g37320*, *LOC_Os11g37390* [22], *LOC_Os09g25060* [4], *Zm00001d038250*, and *Zm00001d001960* [23], have not been reported in mung bean using the GWAS. Therefore, this limits us in the characterization of the genetic basis and molecular mechanism of alkali tolerance in mung beans.

Besides the GWAS, RNA-seq, which can provide precise characterization of gene expression and construct molecular regulatory pathways, has also been applied for the excavation of candidate genes involved in alkali stress responses. A transcriptomics analysis played a role in revealing the alkalinity adaptation strategies of lentils and in identifying the significantly upregulated differently expressed genes (DEGs) involved in abscisic acid (ABA) signaling and secondary metabolite synthesis [24]. Xu et al. [25] identified 12 cross-differentially expressed genes in response to the alkali stress of rapeseed; the major genes that responded specifically in tolerant material were mainly enriched in carbohydrate metabolism, photosynthetic processes, ROS regulation, and the response to salt stress. In recent years, the combination of the GWAS and RNA-seq has proven to be reliable for identifying potential genes related to abiotic stress. Yuan et al. [26] identified 98 candidate genes that were significantly associated with the salt stress response in cotton using a GWAS, among which, 13 were stably differentially expressed, as indicated by RNA-seq. Some other promising candidate genes that are involved in root development under low nitrogen [27], aluminum toxicity stress [28] in rapeseed, responses to low temperature during germination [29], and the seminal root length of seedlings under drought stress [30] in maize have been validated by exploratory analyses that integrated a GWAS and a transcriptome analysis.

In the present study, a panel of 277 mung bean accessions was collected to investigate two traits at the germination stage that were associated with alkali tolerance using a GWAS. In addition, the accessions with a high alkali tolerance when treated with alkali stress were selected to detect DEGs using a transcriptome analysis. This study identified SNPs that are significantly associated with alkali tolerance and candidate genes for uncovering the molecular mechanisms of the alkali tolerance in mung bean.

## 2. Materials and Methods

### 2.1. Plant Materials and Phenotyping for Alkali Stress

In this study, 277 diverse mung bean accessions were collected and preserved in the Key Laboratory of Crop Gene Resources and Germplasm Creation in Northeast China (Jilin Academy of Agricultural Sciences), Ministry of Agriculture and Rural Affairs. This collection of mung bean accessions, including 102 landraces and 175 breeding lines, was selected from 15 provinces in China (Appendix A). A total of 150 healthy seeds with a uniform size from each accession were sterilized in a 5% NaClO solution for 10 min, after which we washed the seeds three times with distilled water. Next, 25 seeds were placed in an 8 cm-diameter Petri dish with filter paper. A 50 mM mixed-alkali (NaHCO_3_:Na_2_CO_3_ with a molar ratio of 9:1) solution [31] was used to simulate a typical alkali stress environment. A total of 25 mL of distilled water or mixed-alkali solution was added as a control or an alkali stress, respectively. Three replicates were included for each treatment. Finally, the Petri dishes were randomly placed in an incubator at 25 °C for 5 d without light. The distilled water or mixed-alkali solution was replaced on the 3rd day.

Here, the criterion for determining germination was a radicle length that was equal to the seed length. We investigated the seeds for germination every day to analyze the germination rate (GR) and germination index (GI). The GR and GI were calculated as follows: GR = (N_5_/25) × 100% and GI = ∑(G_t_/T_t_), where N_5_ represents the number of germinated seeds on the 5th day, G_t_ represents the accumulated number of germinated seeds on day t, and T_t_ is the time corresponding to G_t_ in days [32]. The relative values of the GR (RGR) and GI (RGI) were used to assess the response of mung bean accessions to alkali stress, viz. the values of the seeds under alkali stress divided by the values of the control seeds. 

### 2.2. Whole-Genome Re-Sequencing and GWAS Analysis

A total of 277 mung bean accessions were re-sequenced by an Illumina NovaSeq PE150 platform, and the newly released Jilv 7 [33] was chosen as the high-quality reference genome. According to the criteria (missing data < 10% and minor allele frequency > 5%), the final SNPs used for the GWAS were selected. Admixture software was used to analyze the population structure [34] (http://dalexander.github.io/admixture/index.html, accessed on 22 July 2022). A linkage disequilibrium (LD) analysis was performed using the PopLDdecay software [35] (http://github.com/BGI-shenzhen/PopLDdecay, accessed on 9 January 2023).

Using the GEMMA software [36], a GWAS using a mixed linear model (MLM) [37] was carried out. A quantile–quantile plot (Q-Q plot) and a Manhattan plot were generated using the ggplot2 software [38] and the qqman software [39], respectively. The threshold for the identification of the SNPs that were significantly associated with traits was set to *p* < 1 × 5^−10^ (−log_10_ *p* > 5). Significant SNPs with a physical distance of less than the LD decay distance were treated as the same QTL, and the SNP with the smallest *p*-value was taken as the peak SNP. Bilateral regions with the LD decay distances of the significant SNPs were selected to identify candidate genes.

### 2.3. RNA Sequencing and Data Analysis

Among the 277 mung bean genotypes, 132–346 with an RGR and RGI of 77.02 and 51.01%, respectively, were selected as high-alkali-tolerant accession for the transcriptome analysis. The seeds were treated in the same way as presented in “Section 2.1”. Embryo tissues were sampled after 24 h of treatment, quickly frozen in liquid nitrogen, and stored at −80 °C for RNA extraction. The samples were designated as TM for the control tissues and TMT for the alkali-treated tissues. Each treatment had three biological replicates. The total RNA was extracted using TIANGEN RNAprep Pure Plant Plus Kit (polysaccharide- and polyphenolic-rich). The quality of the total RNA was checked using an Agilent 2100 Bioanalyzer. Library construction was achieved using the NEBNext^®^ Ultra^TM^ RNA Library Prep Kit for Illumina^®^ according to the manufacturer’s recommendations. The library was sequenced on an Illumina NovaSeq 6000 platform. 

The low-quality raw data were filtered by removing reads containing an adapter and ploy-N. The retained high-quality reads were mapped to the mung bean reference genome Jilv 7 [33] using the Hisat2 software (https://daehwankimlab.github.io/hisat2/, accessed on 14 July 2022) [40]. The gene expression levels were normalized by the expected fragments per kilobase per million fragments sequenced (FPKM) [41]. Differential expression analyses between the control and alkali-stressed treatments were performed using DESeq [42] with a pairwise comparison algorithm. Genes with an adjusted *p*-value, an FDR (false discovery rate) < 0.05, and a |log_2_ (fold change)| ≥ 1 were identified as DEGs. To further determine the biological functions of DEGs, gene ontology (GO) enrichment and Kyoto encyclopedia of genes and genomes (KEGG) pathway analyses were implemented using the GOseq R software package [43] and the KOBAS software [44], respectively. A *p*-value corrected by an FDR < 0.05 was set as the significant threshold for both the GO enrichment and KEGG pathway analyses. 

### 2.4. Quantitative Real-Time PCR (qRT-PCR) Analysis

The alkali-tolerant accessions 132–346 were used for the qRT-PCR. The seeds were treated and sampled with the same method described in “Section 2.3”. The total RNA from 6 samples was extracted, followed by the construction of cDNA libraries using the HiScript 1st Strand cDNA Synthesis Kit. The qRT-PCR reaction was run on a LightCycler 480 Ⅱ (Roche, America) using the AceQ qPCR SYBR Green Master Mix (without ROX) according to the manufacturer’s instructions. Specific primers for the target genes were designed by Primer Premier 6 (Premier Biosoft Inc., San Francisco, CA, USA) as listed in Appendix A. *VrActin* (AF143208.1) was used as an intro-reference gene. Relative gene expression levels were calculated according to the 2^−ΔΔCt^ method [45]. Three technical replicates for each biological replicate sample were analyzed.

## 3. Results

### 3.1. Phenotypic Trait Analysis

The GR and GI under control and alkali stress conditions were investigated to assess the alkali tolerance of 277 mung bean accessions during germination. The descriptive statistics for the phenotypic variation are summarized in Appendix A. The mean GR and GI were significantly lower under alkali stress (33.52% and 6.15, respectively) than under normal conditions (95.16% and 20.74, respectively) (Figure 1A). Under control conditions, the GR and GI ranged from 32.00% to 100.00% and 3.39 to 25.00, respectively. Under alkali stress, the GR and GI ranged from 0 to 93.33% and 0 to 19.33, respectively. The CVs (coefficients of variation, %) of the GR and GI were 67.96 and 70.10, respectively, which were both higher than the values for the control. These results indicated that seed germination was significantly impaired by alkali stress.

Moreover, the RGR and RGI were used to evaluate the alkali tolerance of all accessions. The RGR varied from 0 to 93.33% with an average of 34.67%, and the RGI varied from 0 to 82.27% with an average of 28.46% (Table 1). The CVs of the RGR and RGI were both high, at 66.59% and 63.68%, respectively. The statistics of the frequency distribution (Figure 1B) revealed continuous variation in the RGR and RGI, which demonstrated that the RGR and RGI were appropriate for a subsequent association study. Significant positive correlations were detected between the GR and GI, with correlation coefficients of 0.80 and 0.95 under control and alkali stress conditions, respectively (Figure 1C). The RGR was also positively correlated with the RGI, with a correlation coefficient of 0.91. Notably, significant positive correlations were determined among both the GR and GI under alkali stress with the RGR and RGI; all correlation coefficients were not less than 0.90.

### 3.2. Genome-Wide Association Study

Approximately 1.5 Tb of whole-genome sequencing data were generated for 277 mung bean accessions, with an average depth of 9.03×. As a result of SNP filtering, 889,451 SNPs were used in the GWAS analysis. A total of 277 accessions were divided into 11 groups using admixture software for calculating the population structure (Appendix A). The linkage disequilibrium was estimated between all the SNP markers over the 277 accessions; the LD decay plot (Appendix A) showed that the LD in this mung bean panel was high and decayed slowly. Therefore, we used the distance travelled when the r^2^ value dropped to 0.5 as the LD decay distance, viz. about 300 kb.

A GWAS was conducted with 889,451 SNP markers and two related traits (RGR and RGI). Manhattan and Q-Q plots are shown in Figure 2. A total of 19 QTLs containing 32 SNPs that were significantly associated with the RGR and RGI are shown in Table 2. These SNPs were located on chromosomes 1, 2, 3, 4, 6, 7, 8, 10, and 11, with the highest number of SNPs (eight) on chromosomes 4 and 7. The genetic variation explained by these QTLs (R^2^) varied from 3.6 to 14.6%. For the RGR, four QTLs were identified that contained four SNPs distributed on four different chromosomes. For the RGI, 15 QTLs containing 28 SNPs were detected. Two peak SNPs, Chr3_1599285 and Chr6_39567121, excavated in the QTLs *qRGR3* and *qRGI3-1* and the QTLs *qRGR6* and *qRGI6*, respectively, were associated with both the RGR and RGI. The phenotypic contribution (R^2^) of Chr3_1599285 and Chr6_39567121 was 11.3 and 14.6% for the RGR and 11.3 and 13.7% for the RGI, respectively.

The LD decay intervals of the 32 significant SNPs were used to mine candidate genes. In total, 691 genes were identified near the 32 SNPs from the 19 QTLs (Appendix A), but only 312 genes were functionally annotated. Furthermore, we identified 74 candidate genes simultaneously detected in the RGR and the RGI, but only 21 genes were functionally annotated. The GO enrichment analysis (*p*-value < 0.05 and number of genes ≥ 5) indicated that the 312 genes were categorized into 21 GO terms (Appendix A). Among these, macromolecule modification, the phosphate-containing compound metabolic process, the oxidation–reduction process, the cellular protein modification process, and the carbohydrate-derivative metabolic process were enriched in biological processes (BPs), as well as the anaphase-promoting complex in the cellular component (CC). In addition to molecular functions (MFs) related to transferase activity, zinc ion binding, the ubiquitin protein transferase activity, the UDP glycosyltransferase activity, and flavin adenine dinucleotide binding were enriched.

### 3.3. Transcriptome Sequencing Analysis

The transcriptomes of the alkali-tolerant accessions 132–346 under alkali stress and control conditions were analyzed. Consequently, more than 259.55 million clean reads from six libraries were yielded. The average guanine–cytosine (GC) content was 44.69%, and all of the Phred quality scores (Q30s) were not less than 93.08%. On average, more than 93% of the clean reads were mapped uniquely to the reference genome (Appendix A). A total of 2565 DEGs were detected between the alkali stress and control groups 24 h post-treatment, and 1594 (62.14%) were upregulated (Figure 3).

To verify the expression of DEGs using qRT-PCR, 14 DEGs were selected, including eight continuously upregulated and six downregulated genes, and the expression trends were highly similar to those found by RNA-seq, which showed that the transcriptome data were valid (Appendix A).

To further interpret the functional significance of the DEGs, a GO enrichment analysis was implemented. In total, 22 GO terms were enriched (adjusted *p*-value (Padj) < 0.05), including sixteen terms in biological processes (BPs) and six terms in molecular functions (MFs) (Appendix A, Figure 4). For the BP category, the response to oxidative stress, the cell wall organization or biogenesis, the response to stress, the defense response, the aminoglycan catabolic process, the chitin metabolic process, the amino sugar metabolic process, the cell wall macromolecule catabolic and metabolic processes, and the glucosamine-containing compound metabolic process were significantly enriched. The MF category consisted of peroxidase activity, oxidoreductase activity (acting on peroxide as the acceptor), antioxidant activity, heme binding, tetrapyrrole binding, and chitinase activity. In addition, the cell wall, external encapsulating structure, and cell periphery were enriched in the cellular component (CC) (*p*-value < 0.05) (Appendix A, Figure 4). A KEGG analysis was also performed on the DEGs, and eight pathways were significantly enriched (Padj < 0.05), including phenylpropanoid biosynthesis; pentose and glucuronate interconversions; valine, leucine, and isoleucine degradation; tyrosine metabolism; isoquinoline alkaloid biosynthesis; propanoate metabolism; isoflavonoid biosynthesis; and photosynthesis—antenna proteins (Figure 5).

### 3.4. Screening Candidate Genes by Combining Analyses of GWAS and RNA-Seq

The potential alkali tolerance candidate genes were further screened by integrating the GWAS and RNA-seq analyses. A total of 208 and 557 genes in the GWAS were detected in the RGR and RGI, respectively (Appendix A). Among these genes, 18 and 20 candidate genes (Table 3) were detected in the RGR and RGI, respectively, which exhibited significantly different expression levels according to RNA-seq (Figure 6). Among the 74 common genes screened from the two shared-peak SNPs (Appendix A), six were simultaneously detected as DEGs in the RGR and RGI (Figure 6), including those encoding a CBS (cystathionine β synthase) domain-containing protein (*jg7600*), a defensin-like protein (*jg7622*), two EXORDIUM proteins (*jg19402* and *jg19404*), a 7-deoxyloganetin glucosyltransferase (*jg19416*), and an uncharacterized protein (*jg7597*).

To further validate the putative genes, all five candidate genes (*jg19402* was highly homologous to *jg19404*) were analyzed by qRT-PCR. Except for *jg19416*, which was significantly downregulated, the other four genes were significantly upregulated after the alkali stress treatment (Figure 7). The results of the qRT-PCR indicated that they might be involved in alkali tolerance responses.

## 4. Discussion

The early stages of plant growth and development (germination stage and seedling stage) are the most sensitive to salinity–alkalinity [46]. During germination under saline conditions, the salt concentration is higher at the seed depth than at lower levels in the soil profile due to the capillary rise of salts [47]. The saline–alkaline stress on the crop initially decreases the germination rate and inhibits the seedling growth [48]. A low germination rate results in incomplete seedlings, which may reduce the crop yield [49]. Therefore, seed germination is extremely important among the stages of the growth cycle of a crop. The germination rate (GR) and germination index (GI), which can intuitionally reflect the germination capacity, have been widely used to evaluate the stress tolerance of the germplasms of different crops in the germination stage and identify candidate genes using a GWAS [4,26,31,50,51,52,53]. Mung bean is sensitive to salinity–alkalinity during germination [31,53]. In the present research, we investigated the two traits (GR and GI) under alkali stress and control conditions, and used their relative values (RGR and RGI) in a GWAS, which will help us understand the genetic basis of alkali tolerance and improve the alkali tolerance of mung bean.

Here, 19 QTLs containing 32 SNPs that were significantly associated with the RGR and RGI were detected using a GWAS, which explained 3.6 to 14.6% of the phenotypic variation (Table 2). Since the RGR and RGI exhibited strong correlations (Figure 1C), two pleiotropic SNPs were excavated, including Chr3_1599285 and Chr6_39567121, which were significantly associated with both traits. In particular, the SNP Chr6_39567121, with the highest phenotypic contribution of 14.6 and 13.7% for the RGR and RGI, respectively, could be the locus with the most potential for MAS (marker-assisted selection). There have been no previous reports about QTL- or SNP-related alkali tolerance identified in mung bean, so we compared our results to the limited previous studies [19,20] related to salt tolerance. There was no co-location QTL among the QTLs identified in the present study, which may be a result of the complexity of alkali tolerance mechanisms differing from that of salt tolerance in mungbean, and a result of the differences in the reference genomes [33,54,55], populations, and phenotypic traits in the respective studies. 

In view of the negative idea that the GWAS results possibly included false-positive candidate genes [56], we performed an RNA-seq analysis of the alkali-tolerant accession 132–346 under alkali-stress and control conditions after 24 h of treatment. According to the accurate transcriptome data, 2565 DEGs were identified (Figure 3). We combined the results of the RNA-seq and the GWAS, and six DEGs (jg7600, jg7622, jg19402, jg19404, jg19416, and jg7597) (Figure 6) that were simultaneously detected in the RGR and RGI using the GWAS were taken as the hub genes associated with alkali tolerance. 

*Jg7600* is homologous to *Arabidopsis CBSX5* (*At4g27460*), which encodes one of the CBS-domain-containing proteins (CDCPs) (Table 3). In rice, the CDCP genes, which contain a high frequency of stress-responsive *cis*-regulatory elements, are responsive to various abiotic stresses, including salinity [57]. The CBSX family is a subclass of CDCPs, with the basic feature of harboring only a single CBS domain [58]. CBSXs may play a defensive role against various threats, especially those that evoke bursts in reactive oxygen species (ROS) production in the cell, and they enable the cell to cope with life-threatening conditions through redox regulation, embracing all life processes, including seed germination [59,60]. Moreover, *CBSX5* is expressed differentially in both the roots and shoots under various stress conditions, including salt stress [58]. When *OsCBSX4* was transferred into tobacco, it enabled the plants to achieve a better tolerance of salt stress [61]. 

*Jg7622* encodes a defensin-like protein (DEFL) (Table 3), the alternative name of which is clone PSAP10. It is required for seed germination in *Vigna unguiculata* [62]. Defensins and DEFLs, which are cationic cysteine-rich peptides involving the CSαβ motif, belong to antimicrobial peptides (AMPs) and have a role in the adaptability of plant environmental stress [63]. The overexpression of a laminarin-induced *DEFL202* (*At4g11393*) enhanced chloroplast stability under salinity stress and improved the vitality of plant growth after heat stress [64]. Recently, some plant defensin-like proteins focusing on mediating cadmium tolerance or accumulation have been reported, such as *BnaC02g23620D* in *Brassica napus* [65]; *CAL1* [66], *CAL2* [67], and *DEF8* [68] in rice; and *AtPDF2.5* [69] and *AtPDF2.6* [70] in *Arabidopsis*.

Both *jg19402* and *jg19404* are homologous to *Arabidopsis At4g08950*, which encodes the same protein, EXORDIUM (EXO) (Table 3). *EXO* is characterized as a BR (brassinosteroid)-upregulated gene [71], and the counterparts herein were also significantly upregulated after the alkali stress treatment (Figure 7). The EXO protein might mediate BR-induced cell expansion via the modification of cell wall properties and metabolism [72], which is in accordance with the results of the GO enrichment analysis indicating that the terms of cell wall organization or biogenesis, the cell wall macromolecule catabolic and metabolic processes in BP, and the cell wall in CC were significantly enriched (Appendix A, Figure 4). The *Arabidopsis EXORDIUM-LIKE1* (*EXL1*) gene (*At1g35140*) adapted plants to low carbon and oxygen-deprivation conditions [73,74] via balancing the supply and demand to optimize the capacity for sustained growth.

*Jg19416* allowed for the functioning of UDP glucosyltransferase (UGT) activity by encoding a 7-deoxyloganetin glucosyltransferase (UGT85A24) (Table 3). UGTs, which facilitate glycosylation by catalyzing the transfer of uridine 5′-diphospho sugars to specific receptor molecules [75,76], including all major classes of plant secondary metabolites and plant hormones [77], mitigate the effects of oxidative stress under various abiotic stresses [78]. *UGT85A5* [75], *UGT84A4* [79], *CrUGT87A1* [80], and *PhUGT51* [81] were identified as playing a role in the regulation of salt resistance, of which *CrUGT87A1* is involved in salt tolerance as a result of increasing flavonoid accumulation [80]. In addition, *AtUGT79B2* and *AtUGT79B3* could improve the tolerance to chilling damage through increasing the anthocyanin accumulation [82], and *NtUGT127* and *NtUGT245* might respond to drought stress by modulating the flavonoid content [76]. In the present research, pathways of secondary metabolite biosynthesis (phenylpropanoid, isoquinoline alkaloid, and isoflavonoid) were significantly enriched in the KEGG analysis (Figure 5); we supposed that these secondary metabolites might be the substrates of UGT85A24, which could be related to the regulation of alkali tolerance.

Proteins of unknown function were suggested to be the contributing factors for various physiological and biochemical processes [83]. Therefore, understanding the functional characteristics of these proteins would allow for the discovery of novel or alternate pathways [61], such as the last gene here, *jg7595* (Table 3). Moreover, the other 26 potential alkali tolerance candidate genes (Table 3) that were screened only in the RGR or RGI by combining the analysis of the GWAS and RNA-seq should also be further studied.

## 5. Conclusions

The alkali tolerance of 277 mung bean accessions was evaluated with the relative values of two germination traits, and a GWAS for alkali tolerance was conducted. A total of 19 QTLs containing 32 SNPs that were significantly associated with the two germination traits were identified on nine chromosomes of mung bean, and 691 candidate genes were mined within the SNP flanking sequences. Transcriptome sequencing of the alkali-tolerant accession 132–346 under alkali and control conditions after 24 h of treatment was performed, and 2565 DEGs were identified. The candidate genes and the DEGs were integrated, six hub genes were selected, and the expression of these genes was further validated using qRT-PCR. These findings provide a solid foundation for revealing the molecular mechanism of alkali tolerance in mung bean, and provide a chance to develop alkali-tolerant varieties through MAS.

## Figures and Tables

**Figure 1 genes-14-01294-f001:**
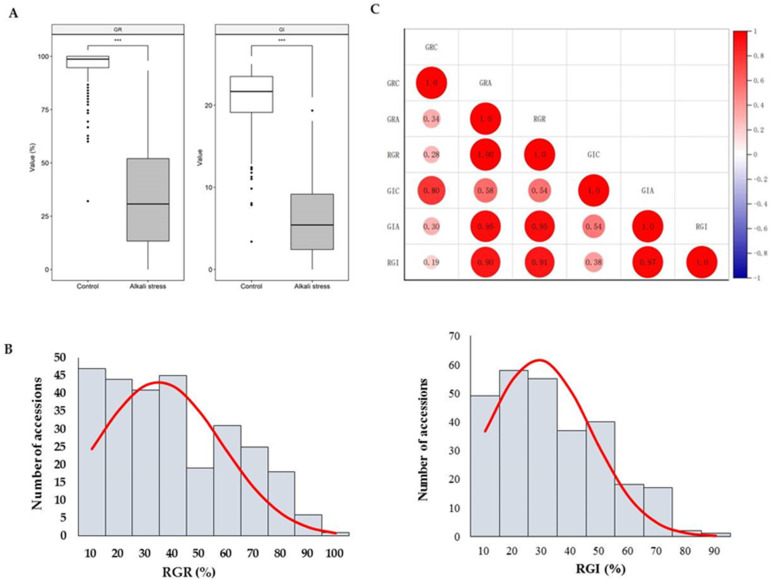
Phenotypic analysis of germination-related traits under control and alkali stress conditions. (**A**) Distribution of gemination rate (GR) and germination index (GI) under control and alkali stress; (**B**) frequency distribution of the RGR and RGI for the 277 mung bean accessions; and (**C**) correlation analysis of alkali-tolerance-related traits. GRC: germination rate under control conditions; GRA: germination rate under alkali stress; RGR: relative germination rate; GIC: germination index under control conditions; GIA: germination index under alkali stress; RGI: relative germination index. *** indicates a significant difference at *p* < 0.001.

**Figure 2 genes-14-01294-f002:**
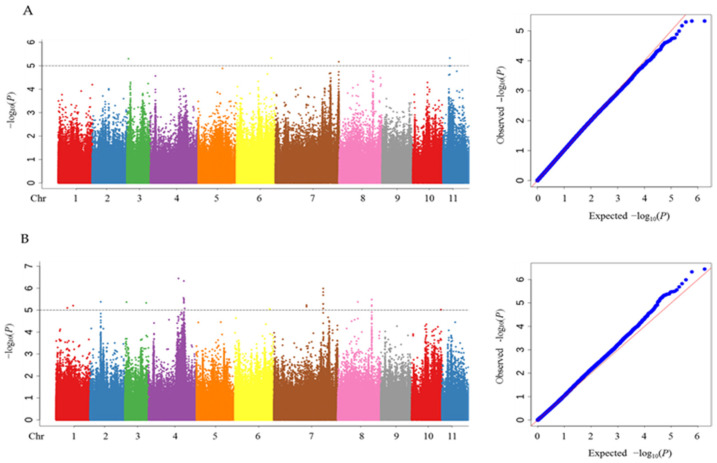
Manhattan and Q-Q plots of GWAS for (**A**) RGR and (**B**) RGI. The horizontal dotted lines represent the thresholds for identifying significant SNPs.

**Figure 3 genes-14-01294-f003:**
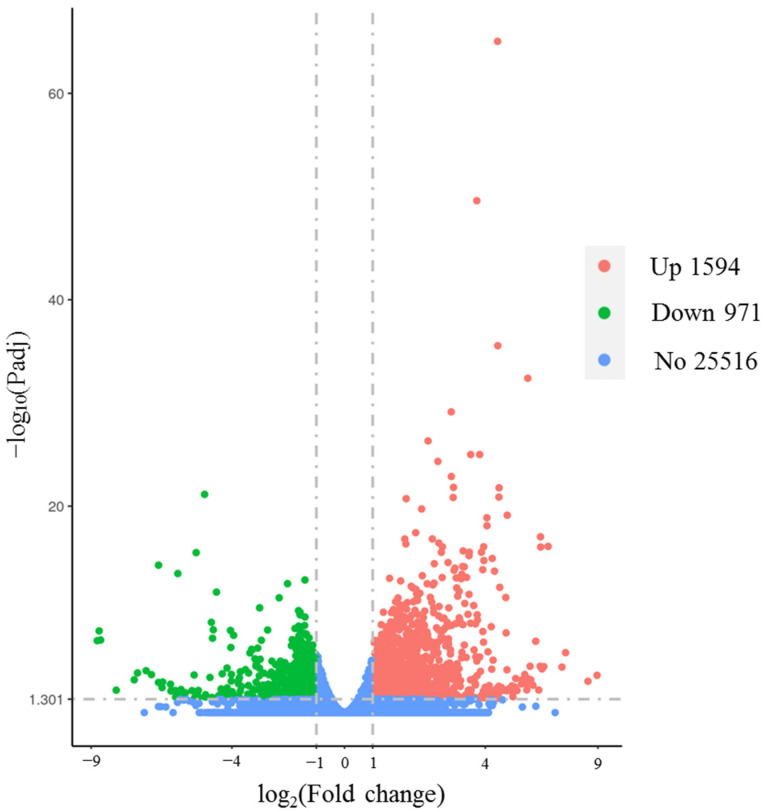
Volcano plot displaying DEGs of alkali-tolerant accession 132–346 under alkali stress and control conditions. Padj represents the adjusted *p*-value; Padj < 0.05 and |log2 (fold change)| ≥ 1.

**Figure 4 genes-14-01294-f004:**
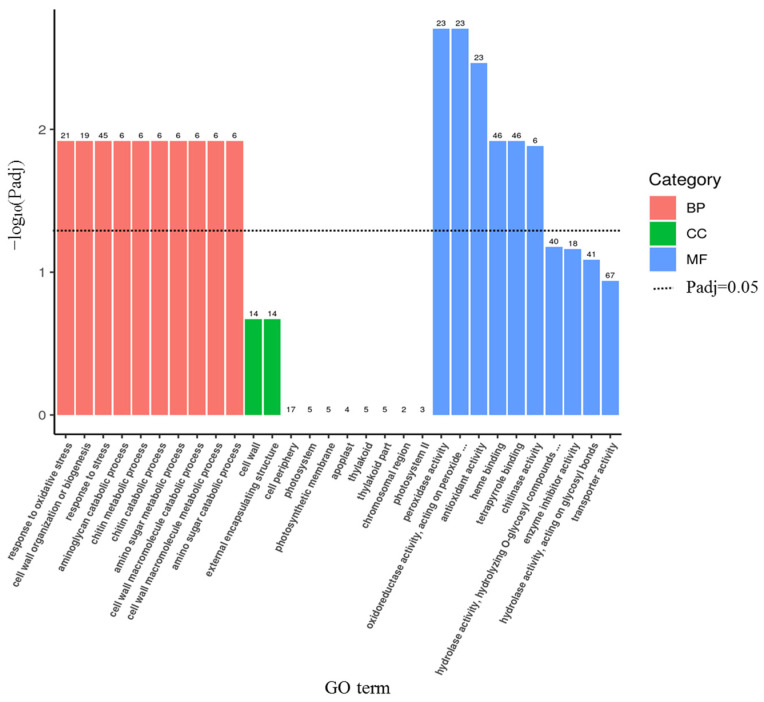
GO functional enrichment analysis of DEGs identified by RNA-seq. Padj represents the adjusted *p*-value. The number on each bar is the number of DEGs enriched in the corresponding GO term.

**Figure 5 genes-14-01294-f005:**
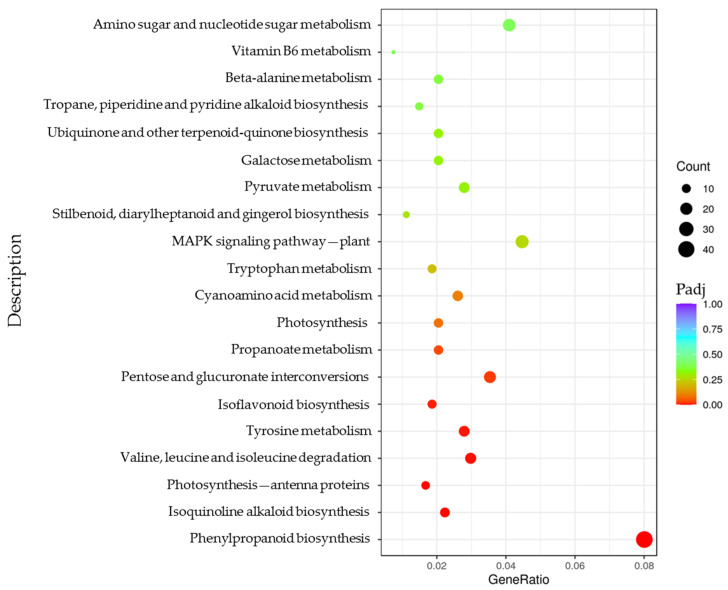
KEGG enrichment analysis of DEGs detected by RNA-seq. Padj represents the adjusted *p*-value.

**Figure 6 genes-14-01294-f006:**
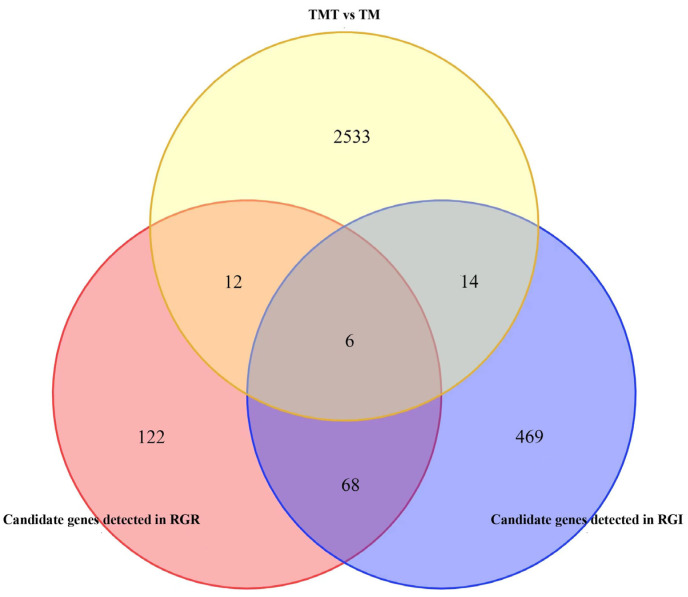
Venn diagram of DEGs identified by RNA-seq and candidate genes detected by GWAS in RGR and RGI.

**Figure 7 genes-14-01294-f007:**
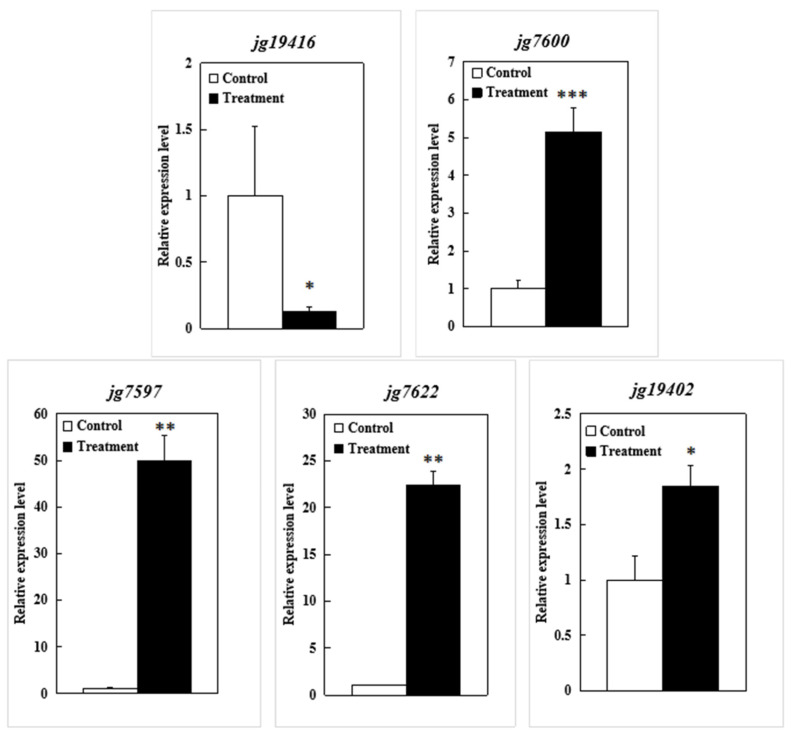
qRT-PCR validation for putative genes of alkali stress tolerance screened by combining the analyses of GWAS and RNA-seq. *, **, and *** indicate significant differences at *p* < 0.05, *p* < 0.01, and *p* < 0.001, respectively.

**Table 1 genes-14-01294-t001:** Basic statistics of RGR and RGI in the 277 mung bean genotypes under control and alkali stress environments.

Traits	Min (%)	Median (%)	Mean (%)	SD	Skewness	Kurtosis	CV (%)
RGR	0.00	32.00	34.67	23.09	0.43	−0.85	66.59
RGI	0.00	26.35	28.46	18.12	0.52	−0.57	63.68

RGR: relative germination rate; RGI: relative germination index; Min: minimum; Max: maximum; SD: standard deviation; CV: coefficient of variation.

**Table 2 genes-14-01294-t002:** QTLs and SNPs related to alkali tolerance detected by GWAS.

Traits	QTLs	No. of SNPs	Peak SNP	Chr.	Position	Alleles	−log_10_P	R^2^ (%)
RGR	*qRGR3*	1	Chr3_1599285	3	1599285	T/C	5.30	11.3
	*qRGR6*	1	Chr6_39567121	6	39567121	G/A	5.33	14.6
	*qRGR7*	1	Chr7_72278080	7	72278080	G/A	5.17	4.0
	*qRGR11*	1	Chr11_7640538	11	7640538	G/A	5.33	3.6
RGI	*qRGI1-1*	1	Chr1_12682868	1	12682868	C/A	5.10	13.4
	*qRGI1-2*	1	Chr1_19279047	1	19279047	C/T	5.20	8.9
	*qRGI2*	1	Chr2_11936562	2	11936562	T/C	5.38	5.0
	*qRGI3-1*	1	Chr3_1599285	3	1599285	T/C	5.37	11.3
	*qRGI3-2*	1	Chr3_24016456	3	24016456	A/C	5.33	10.1
	*qRGI4-1*	1	Chr4_34059117	4	34059117	T/A	6.44	5.1
	*qRGI4-2*	1	Chr4_39812897	4	39812897	G/A	5.56	13.6
	*qRGI4-3*	5	Chr4_40188515	4	40188515	C/G	6.33	4.8
	*qRGI4-4*	1	Chr4_41058948	4	41058948	A/G	5.08	4.3
	*qRGI6*	1	Chr6_39567121	6	39567121	G/A	5.05	13.7
	*qRGI7-1*	2	Chr7_37173579	7	37173579	G/A	5.23	11.9
	*qRGI7-2*	6	Chr7_56140780	7	56140780	C/G	5.99	8.4
	*qRGI8-1*	1	Chr8_22721042	8	22721042	A/G	5.37	12.3
	*qRGI8-2*	4	Chr8_38508969	8	38508969	G/A	5.48	4.2
	*qRGI10*	1	Chr10_33340353	10	33340353	C/T	5.02	12.5

R^2^ represents the phenotypic variance explained by the QTL.

**Table 3 genes-14-01294-t003:** The candidate genes identified by integrating GWAS and RNA-seq.

Gene ID	Gene Name	TMT vs. TM	Description	GO in UniProtKB
*jg7597*	*NA*	up	NA	NA
*jg7600*	*CBSX5*	up	CBS domain-containing protein	AMP binding, protein kinase binding, protein kinase regulator activity, cellular response to glucose starvation, cellular response to hypoxia, protein phosphorylation, and regulation of catalytic activity
*jg7622*	*NA*	up	Defensin-like protein	Defense response to fungi and killing of cells of another organism
*jg19402*	*EXO*	up	Protein EXORDIUM	Response to brassinosteroid
*jg19404*	*EXO*	up	Protein EXORDIUM	Response to brassinosteroid
*jg19416*	*UGT85A24*	down	7-Deoxyloganetin glucosyltransferase	UDP glucosyltransferase activity
*jg25742*	*MFT*	up	Protein MOTHER of FT and TFL1	Abscisic-acid-activated signaling pathway, positive regulation of seed germination, and response to abscisic acid
*jg25753*	*FKBP17-2*	up	Peptidyl-prolyl cis-trans isomerase	Peptidyl-prolyl cis-trans isomerase activity
*jg25764*	*At2g01390/At2g01380*	down	Pentatricopeptide repeat-containing protein	NA
*jg25771*	*NMT1*	down	Phosphoethanolamine N-methyltransferase 1	Methyltransferase activity, phosphoethanolamine N-methyltransferase activity, choline biosynthetic process, and post-embryonic root development
*jg25787*	*NFXL1*	up	NF-X1-type zinc finger protein	Regulation of hydrogen peroxide metabolism, response to salt stress, salicylic acid biosynthetic process, DNA-binding transcription factor activity, RNA polymerase II-specific activity, and protein ubiquitination
*jg25790*	*NA*	down	Retrovirus-related Pol polyprotein from transposon TNT 1-94	Aspartic-type endopeptidase activity, endonuclease activity, nucleic acid binding, RNA-directed DNA polymerase activity, zinc ion binding, DNA integration, and proteolysis
*jg25802*	*NA*	down	NA	NA
*jg25807*	*AGD15*	down	Probable ADP-ribosylation factor GTPase-activating protein	GTPase activator activity and metal ion binding
*jg34113*	*AVT3C*	up	Amino acid transporter	L-amino acid transmembrane transporter activity and neutral amino acid transmembrane transporter activity
*jg34122*	*MLO8*	down	MLO-like protein 8	Calmodulin binding and defense response
*jg34124*	*ATL8*	up	RING-H2 finger protein	Metal ion binding, transferase activity, and protein ubiquitination
*jg34162*	*At4g19185*	up	WAT1-related protein	Transmembrane transporter activity
*jg14901*	*RPL9*	down	Ribosomal protein L9	Structural constituent of ribosomes, rRNA binding, and translation
*jg14094*	*ACA12*	up	Calcium-transporting ATPase 12; plasma membrane-type	ATPase-coupled cation transmembrane transporter activity, ATP binding, ATP hydrolysis activity, calmodulin binding, metal ion binding, and P-type calcium transporter activity
*jg14117*	*AVT6A*	up	Amino acid transporter	L-amino acid transmembrane transporter activity
*jg10665*	*PSKR2*	up	Phytosulfokine receptor 2	ATP binding, peptide receptor activity, protein serine/threonine kinase activity, protein serine kinase activity, and protein phosphorylation
*jg5709*	*KTI3*	up	Trypsin inhibitor A	Serine-type endopeptidase inhibitor activity
*jg5712*	*NA*	up	NA	NA
*jg6099*	*NA*	up	NA	NA
*jg37480*	*NA*	down	NA	NA
*jg37481*	*SMC1*	down	Structural maintenance of chromosome protein 1	ATP binding, ATP hydrolysis activity, DNA binding, cell division, chromosome segregation, and DNA repair
*jg38807*	*ERF1B*	up	Ethylene-responsive transcription factor 1B	DNA-binding transcription factor activity, defense response, ethylene-activated signaling pathway, and jasmonic-acid-mediated signaling pathway
*jg38809*	*ERF096*	up	Ethylene-responsive transcription factor	DNA-binding transcription factor activity, ethylene-activated signaling pathway, positive regulation of abscisic-acid-activated signaling pathway, and positive regulation of cellular defense response
*jg38828*	*TRAF1B*	up	TNF-receptor-associated factor homolog 1b	Autophagosome organization and innate immune response
*jg38837*	*NA*	down	NA	NA
*jg32966*	*RZF1*	up	E3 ubiquitin protein ligase	Metal ion binding, ubiquitin protein ligase activity, ubiquitin protein transferase activity, negative regulation of proline biosynthetic process, regulation of response to osmotic stress, regulation of response to water deprivation, response to osmotic stress, response to water deprivation, and water homeostasis

## Data Availability

Not applicable.

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
