# Peer review of "Integration of GWAS and RNA-Seq Analysis to Identify SNPs and Candidate Genes Associated with Alkali Stress Tolerance at the Germination Stage in Mung Bean"

_genes, 2023, doi:10.3390/genes14061294_

Round 1
Reviewer 1 Report
The manuscript entitled "Integration of GWAS and RNA-seq Analysis to Identify SNPs and Candidate Genes Associated with Alkali Stress Tolerance at the Germination Stage in Mungbean" reported the research of using 277 mungbean genotypes to identify genomic regions associated with alkali stress tolerance, which identified 19 QTLs and 691 candidate genes from corresponding LD blocks. Combining RNA-seq in one line with GWAS results, 6 hub genes were detected. Overall, the research was conducted in a logic way, and obtained data from the research strongly supported the conclusion. The manuscript is prepared well, few minor issues may need to consider before accepting for publication.
1) The experiment seems used 5-day interval for germination data. Did authors keep observing germination after 5 days? Some seeds may have stronger dormancy and need more days for germinating. If any seeds germinated after 5 days, did that included into the total germination?
2) How about dormancy of the mungbean seeds?
3) How can you conclude the alkali stress tolerance simply based on the germination under alkali condition? Why not the seedlings grown under alkali condition? Or even adult plants grown in the alkali condition? How much the germination under alkali condition is related to tolerance in seedling or adult plants stage?
4) The real field condition will not be consistently as the condition in a growth chamber, how much confidence to utilize the results from the controlled condition for the reality of planting seeds in alkali field? Seed germination is also affected by temperature, if germination tests under different temperature were conducted, this research will be much better.
5) RNA-seq used only one genotype is a significant weakness of this research. If more than two genotypes were used, DEGs commonly detected in multiple genotypes will be more reliable.
6) Overall, the results from this research are still a bit rough for understanding molecular base of alkali stress tolerance, could be a preliminary investigation of genetic mechanism of mungbean tolerance to alkali stress.
7) Why authors chose LOD=5 as a threshold?
8) QQ-plot of RGR in Figure 2 indicates the data quality is low.
9) The alignment in Tables 2 and 3 need to change to upright to make the table easier to read. For example, the traits column in Table 2 is aligned at middle, all columns in table 3 are aligned at middle.
English seems good, but the format of tables need to change as mentioned above.
Reviewer 2 Report
The article "Integration of GWAS and RNA-seq Analysis to Identify SNPs 2 and Candidate Genes Associated with Alkali Stress Tolerance 3 at the Germination Stage in Mungbean" is overall very sound.
First of all however please upload all sequencing data to NCBI, including the raw reads for RNAseq and genome sequencing.
The authors introduce the evolution of stress response, but should flesh this out a bit that this hinges on conserved patterns that have been pivotal throughout the evolutionary history of land plants, see and cite: Evo-physio: on stress responses and the earliest land plants, Journal of Experimental Botany, Volume 71, Issue 11, 11 June 2020, Pages 3254–3269, https://doi.org/10.1093/jxb/eraa007
This also includes the mentioned metabolic responses. Crossroads in the evolution of plant specialized metabolism, Seminars in Cell & Developmental Biology, Volume 134, 30 January 2023, Pages 37-58, https://doi.org/10.1016/j.semcdb.2022.03.004
On Figure 6: Since these are so few in the overlaps, maybe clearly wright next to it what they are (e.g. the 6)
Figure 7: please plot a jitter plot of the individual data points on top of the bar diagrams so that we can see the distribution of the data
I found a few grammar mistakes
